# Effects of Coffee and Its Components on the Gastrointestinal Tract and the Brain–Gut Axis

**DOI:** 10.3390/nu13010088

**Published:** 2020-12-29

**Authors:** Amaia Iriondo-DeHond, José Antonio Uranga, Maria Dolores del Castillo, Raquel Abalo

**Affiliations:** 1Food Bioscience Group, Department of Bioactivity and Food Analysis, Instituto de Investigación en Ciencias de la Alimentación (CIAL) (CSIC-UAM), Calle Nicolás Cabrera, 9, 28049 Madrid, Spain; amaia.iriondo@csic.es (A.I.-D.); mdolores.delcastillo@csic.es (M.D.d.C.); 2High Performance Research Group in Physiopathology and Pharmacology of the Digestive System NeuGut-URJC, Department of Basic Health Sciences, Faculty of Health Sciences, Campus de Alcorcón, Universidad Rey Juan Carlos (URJC), Avda. de Atenas s/n, 28022 Madrid, Spain; jose.uranga@urjc.es; 3Associated Unit to Institute of Medicinal Chemistry (Unidad Asociada I+D+i del Instituto de Química Médica, IQM), Spanish National Research Council (Consejo Superior de Investigaciones Científicas, CSIC), 28006 Madrid, Spain

**Keywords:** brain–gut axis, caffeine, coffee, coffee by-products, dietary fiber, enteric, gastrointestinal, melanoidins, mucosa, myenteric

## Abstract

Coffee is one of the most popular beverages consumed worldwide. Roasted coffee is a complex mixture of thousands of bioactive compounds, and some of them have numerous potential health-promoting properties that have been extensively studied in the cardiovascular and central nervous systems, with relatively much less attention given to other body systems, such as the gastrointestinal tract and its particular connection with the brain, known as the brain–gut axis. This narrative review provides an overview of the effect of coffee brew; its by-products; and its components on the gastrointestinal mucosa (mainly involved in permeability, secretion, and proliferation), the neural and non-neural components of the gut wall responsible for its motor function, and the brain–gut axis. Despite in vitro, in vivo, and epidemiological studies having shown that coffee may exert multiple effects on the digestive tract, including antioxidant, anti-inflammatory, and antiproliferative effects on the mucosa, and pro-motility effects on the external muscle layers, much is still surprisingly unknown. Further studies are needed to understand the mechanisms of action of certain health-promoting properties of coffee on the gastrointestinal tract and to transfer this knowledge to the industry to develop functional foods to improve the gastrointestinal and brain–gut axis health.

## 1. Introduction

In the past years, coffee has gone from being the villain in the movie to the paradoxical hero. In 1991, the International Agency for Research on Cancer (IARC), the specialized cancer agency of the World Health Organization (WHO), classified coffee as “possibly carcinogenic to humans” (Group 2B). This assessment was made on the basis of limited evidence on the association of urinary bladder cancer and coffee consumption. In 2016, after a re-evaluation based on more than 1000 observational and experimental studies, 23 scientists from 10 different countries concluded that the extensive scientific literature does not show evidence of an association between coffee consumption and cancer [1]. Therefore, coffee was moved from Group 2B (“possibly carcinogenic to humans”) to Group 3 (“not classifiable as to carcinogenicity”). In addition, the IARC found that there is evidence that coffee consumption may actually help reduce occurrence of certain cancers (colon, prostate, endometrial, melanoma, and liver) [1,2].

The “coffee paradox” consists of the fact that caffeine raises blood pressure, but drinking coffee is associated with a lower risk of hypertension [3]. In fact, daily coffee consumption is associated with a decrease in the prevalence of heart attack, despite the tendency to smoke in coffee drinkers [4]. In addition, moderate consumption of 3–4 cups of coffee a day is associated with greater longevity and lower risk of all-cause mortality [5]. Coffee consumption has also evidence-based beneficial associations with metabolic diseases (type 2 diabetes, metabolic syndrome, renal stones, and different liver conditions) and neurodegenerative diseases (Parkinson’s and Alzheimer’s disease) [2]. Therefore, coffee consumption is recommended as part of a healthy diet [6,7], since it contains several bioactive compounds with therapeutic properties [8].

Table 1 shows the chemical composition of green, roasted, and brewed coffee. The composition of the green coffee bean is severely affected by the roasting process, during which, among others, the Maillard reaction occurs. This reaction reduces the amount of free chlorogenic acids (CGAs), but other antioxidant compounds are formed, such as melanoidins that incorporate CGA to their backbone (Table 1) [9]. These compounds, among others formed during processing, are responsible for the brown color of roasted coffee beans and contribute to the antioxidant capacity of coffee [10]. On the other hand, the Maillard reaction produces newly formed contaminants, such as acrylamide. The European Commission indicates that levels of acrylamide in coffee can be lowered by the following mitigation approaches: controlling roasting conditions or treating with asparaginase [11]. Roasted coffee is a complex mixture of thousands of bioactive compounds, and some of them have potential health-promoting properties such as antioxidant, anti-inflammatory, antifibrotic, or antiproliferative effects [5].

The brewing procedure will also have an influence on the biochemical composition of the final coffee cup (Table 1) [30]. Coffee brewing is a solid–liquid extraction that involves water absorption by ground roasted coffee, solubilization of ground coffee in hot water, and separation of the water extract from spent coffee grounds. Many variables will affect the composition of the coffee cup, such as coffee particle size, time of extraction, pressure, type of filter, and water temperature, among others [31]. In the last years, consumers have shown great interest in “cold brew”, a coffee beverage prepared with cold water (room temperature or refrigerated water) for up to 24 h [32]. Recent studies indicate that hot and cold brew coffees have small but important differences, particularly in the total antioxidant capacity of the resulting coffee [21]. Although melanoidins have not been characterized in cold brew coffee (Table 1) [33], water extraction temperature causes differential solubility of these molecules [34]. Therefore, further studies are needed to complete the chemical characterization of this popular beverage.

Whatever the brewing method, coffee and its components exert profound effects on the body, and some have already been mentioned above. As for any other food or beverage, the gastrointestinal tract is the first body system that gets in contact with coffee, and local effects do occur. Of course, other gastrointestinal effects occur after absorption of the different coffee components, with these also worth mentioning. Thus, the first part of this review focuses on the effects that coffee, its by-products and its components have been demonstrated to produce in the gastrointestinal tract. These may affect the function of different components of the gut wall (namely, mucosa, muscle, and intrinsic innervation), along the different organs of the gastrointestinal tract (Figure 1), and therefore their effects related with gastrointestinal cancer, inflammation, and mucosa functions (permeability, secretion) as well as in motor function will be discussed.

Furthermore, the gastrointestinal tract is functionally connected with the brain, through the so-called brain–gut axis (or gut–brain axis) [35]. Whereas the effects of coffee and its components on the brain have been deeply studied, those on the brain–gut axis have received comparatively little attention. However, a large amount of evidence has accumulated on the association between psychological factors and gut sensory, motor, and immune functioning [36]. Accordingly, it is now recognized that a healthy brain–gut axis is key for emotional and affective stability, adequate responses to stress, and visceral pain modulation [37]. In fact, the increased awareness about the importance of the brain–gut interaction in gastrointestinal disorders has even given rise to the field of psychogastroenterology [38,39]. Thus, this review also briefly describes the effects that coffee, its by-products and its components have on the brain–gut axis and their possible role in this area.

## 2. Coffee and the Gastrointestinal Tract: Focus on the Mucosa

The effects of coffee on the gastrointestinal tract have been studied for years in order to understand its hypothetical stimulating or inhibiting properties and its mechanisms of action. This problem has been addressed through numerous epidemiological studies. These works have been mainly focused on neoplastic diseases, with conflicting results, although there is evidence suggesting that coffee may be associated with lower risk of some cancers. Indeed, systematic reviews have found a protective effect of coffee on liver, hepatocellular, and breast cancers. However, coffee seems to increase the risk for lung cancer development, whereas the association of coffee with other cancers such as those of the pancreas, bladder, ovaries, and prostate is controversial [40,41]. Regarding cancers of the digestive tract, most meta-analyses have revealed a modest or dose–response-inverse association between coffee and the risk of colorectal cancer (CRC) [42,43,44,45,46,47,48]. In particular, coffee consumption has been found to be inversely associated with risk of CRC in a dose-dependent manner in the northern regions of Israel [49] or among Japanese women [50]. Furthermore, a recent prospective observational study including 1171 patients, most of them with metastatic CRC, showed an increase in survival up to 8 months for those who consumed daily four or more cups of coffee [51]. Differences related with race or sex seem to be important when assessing results that can sometimes be conflicting. Thus, a meta-analysis by Micek et al. (2019) [52] did not find any evidence for the association between coffee intake and CRC risk but, when using pooled groups, coffee consumption was related with a decreased risk of colon cancer in never-smokers and in Asian countries, and with an increased risk of rectal cancer in the general population, not considering women, never-smokers, and European countries. Similarly, a systematic review and meta-analysis of 24 prospective studies on CRC showed that coffee exerts a protective effect in men and women combined and in men alone. Regarding ethnicity, a significant protective association was noted in European men and in Asian women. Decaffeinated coffee exhibited a protective effect in both men and women [53]. On the contrary, other researchers have found no protective evidence for coffee. It is worth mentioning the EPIC cohort study by Dik et al. (2014) [54] that involved more than 400,000 Europeans and showed no association between coffee consumption and CRC. Park et al. (2018) [55] also found no relationship between CRC and coffee consumption in a large prospective multiethnic cohort study involving 4096 patients. Similarly, prospective studies among Swedish women did not find any relationship between CRC and the intake of four or more cups of coffee per day [56]. The same type of study among the British population also found no relationship between coffee and stomach, small bowel, or colorectal cancers [57]. In this respect, the results of gastric cancer are difficult to evaluate. Some meta-analyses affirm that coffee diminishes the risk of gastric cancer [58] but in other cases the results are conflicting, depending directly on the sex of the patients [59,60] or on the part of the stomach studied, with a direct relationship being found between coffee intake and gastric cardia cancer but not other cancers affecting the stomach [61]. Similarly, the association with esophageal cancer does not seem clear since there are systematic reviews that affirm that the relationship between coffee consumption and the incidence of this cancer either does not exist [61,62] or is attributable to the temperature of the beverage [63]. On the contrary, a meta-analysis comparing coffee and tea found a significant correlation between coffee and esophageal cancer [64].

Results of epidemiologic studies on non-neoplastic pathologies are also controversial. Some meta-analyses have shown that overall coffee did not seem to be a causal factor for chronic gastroesophageal reflux disease (GERD) [65], whereas an Italian study found an adverse effect of coffee among Barrett’s esophagus (BE) patients [66]. On the contrary, a survey in the United States did not find any association between coffee intake and the risk of BE [67].

The variability described above may be due to many causes, including sex, ethnicity, lifestyles, and the numerous bioactive compounds present in coffee. In fact, it soon became evident that caffeine, considered the main component of coffee, was not really the only bioactive compound in coffee. In particular, the discovery during the second half of the last century, that even decaffeinated coffee causes an increase in gastric acid secretion and a reduction in the competence of the lower esophageal sphincter [68,69], led to an investigation of the physiological effect of the other coffee derived-compounds. As mentioned above, coffee composition depends on many factors such as coffee origin, method of preparation (water steam temperature, roasting, etc), resulting in variable effects on physiology and microbiome [41,70,71,72,73].

Thus, studies conducted in vivo with animal models or volunteers, or in vitro with isolated cells to separately evaluate the effects of the various compounds present in coffee, are much less numerous than the epidemiological reports. The interspecific differences in metabolism or the different doses tested have a great impact on the results obtained. However, although still incomplete and somehow leading to contradictory results, the efforts to investigate the mechanisms by which coffee manifests its effects and the specific compounds responsible for them have already shed some light on the subject, as shown next.

### 2.1. In Vitro Studies

#### 2.1.1. Coffee

Since the decade of 1980, several studies have investigated whether coffee or its derivatives have carcinogenic effects. These studies identified potentially harmful compounds such as hydrogen peroxide (H_2_O_2_) in various coffee preparations. However, these studies were carried out in bacterial models that lacked the enzymes of peroxisomes, and thus this supposed carcinogenic effect could not apply to humans. The compounds in coffee that are responsible for the generation of these potential harmful effects were not identified either [74,75].

Similarly, the anti-inflammatory properties have been investigated in coffee preparations such as coffee “charcoal”, a herbal preparation produced by roasting green dried coffee and milling to powder. In this case, the intestinal cells increased their barrier function, and inflammatory mediators such as interleukin (IL) IL-6, IL-8, tumor necrosis factor (TNF), methyl-accepting chemotaxis protein-1 (MCP-1), and prostaglandin (PG) E2 were inhibited [76]. However, this preparation also preserves most of the compounds in coffee and it is difficult to identify a specific molecule responsible for these effects.

In line, the incubation of CaCo2 cells (a human colorectal adenocarcinoma cell line), with regular, filtered, decaffeinated, or instant coffee resulted in an induction of the transcription of uridine diphosphate (UDP) glucuronosyltransferases (UGT1A), proteins with indirect antioxidant properties. The responsible molecule for this upregulation remained elusive also in this case [77].

#### 2.1.2. Caffeine

The alkaloid caffeine is amongst the most studied coffee components [41]. Caffeine has been considered to possess antioxidant properties, although very high doses were needed to demonstrate them [78]. On the contrary, when physiological concentrations were used, caffeine did not show any antioxidant activity measured by oxygen-radical absorbing capacity. However, the antioxidant activity was significant when using 1-methylxanthine and 1-methyluric acid, the main metabolites of caffeine in humans. The antioxidant effects of these compounds are equivalent to those produced by ascorbic acid and uric acid, respectively [79]. This does not rule out, however, the involvement of other mechanisms.

Colonic cell lines have also been used to assess the anti-inflammatory activity of caffeine. Co-cultures of human colorectal adenocarcinoma cell line CaCo2 and 3T3-L1 adipocytes in the presence of caffeine have shown that caffeine inhibits the secretion of inflammatory cytokines interleukin (IL) IL-8 and plasminogen activator inhibitor-1 (PAI-1) and decreases lipid accumulation in adipocytes, whereas it has no effect on 3T3-L1 cells alone [80]. Related to this, CaCo2, goblet, and macrophage cell lines have also been co-cultured to study the effects on mechanisms relevant for inflammatory bowel disease (IBD).

In fact, more recent studies focused on caffeine tend to show the opposite. Moreover, caffeine has also been shown to increase sensitivity to radiotherapy of RKO cells in the transition from G1 to G2 phases in the cell cycle [81]. Caffeine may also act synergistically with the suppressor gene phosphatase and tensin homolog (PTEN), suppressing cell growth and inducing apoptosis in several human CRC cell lines but not in fibroblasts. This effect was induced through downregulation of the serine/threonine kinase (AKT) kinase pathway, and modulation of the p44/42MAPK pathway even in the absence of p53 [82]. Additionally, caffeine inhibits hypoxia-inducible factor-1 (HIF-1) in HT29 CRC cells cultured in hypoxic conditions. It also reduces vascular endothelial growth factor (VEGF) promoter activity and IL-8 expression, essential factors involved in tumor angiogenesis. The inhibition of kinases such as extracellular signal-regulated kinase (ERK1/2), *p38*, and AKT through blockade of their phosphorylation might also be the mechanism elicited by caffeine in this case. Additionally, it also inhibited cell migration stimulated by the adenosine A3 receptor [83]. These effects of caffeine may be different in cells of different origin or when it is administered synergically with other molecules. Thus, caffeine cannot inhibit ERK phosphorylation and the consequent epidermal growth factor (EGF)- and H-Ras-induced neoplastic transformation in the JB6 P epithelial cell line [84]. Similarly, caffeine activates the ERK signaling pathway in the Colo-205 CRC cell line, resulting in an increase of the anti-apoptotic protein myeloid cell leukemia 1 (Mcl-1) and a higher resistance to paclitaxel [85]. This effect was not observed with the HT-29 cell line, although in this case incubation with caffeine lasted only 20 h [86]. These differences can be explained considering particularities in cell lines, exposure time, and/or concentration of caffeine assayed. It is also important to consider that in vitro research may not fully reflect the complex relationships in multicellular organisms nor the dosage finally reaching the different tissues in vivo. Regarding this, Guertin et al. (2015) [87] studied a large number of serum metabolites present in coffee drinkers and found that some caffeine-related metabolites were inversely associated with CRC. Experimental in vivo studies are needed to understand the mechanisms underlying the exact caffeine–cancer association.

#### 2.1.3. Polyphenols

Polyphenols are other important compounds present in coffee. They include different concentrations of CGAs, composed of quinic acid with trans-cinnamic acids, with the caffeoylquinic acids (CQAs), especially the 5-O-caffeoylquinic acid (5-CQA) and one of the metabolites of CGA, caffeic acid (CA), as the most studied [70,73]. Polyphenols have strong antioxidant properties both in decaffeinated and regular coffee and may also reduce the activation of proinflammatory factors such as nuclear factor-kβ (NF-kβ) in cultured myoblasts proportionally to CGA concentration, with regular coffee being twice as potent as decaffeinated coffee [88]. Similarly, Zhao et al. (2008) [89] demonstrated that the secretion of IL-8 induced by H_2_O_2_ or by tumor necrosis factor-receptor (TNF-R) activation may be blocked by 5-CQA in a dose-dependent manner in human intestinal epithelial CaCo2 cells. These effects are interesting since over-expression of pro-inflammatory elements and increased amounts of reactive oxygen species (ROS) are closely associated with DNA damage and multiple cell-signaling pathways involved in pathogenesis of important diseases such as cancer [41,90]. Moreover, 5-CQA and especially CA inhibit cell growth in the transition from G1 to G2/M phases of the cell cycle in the HT-29 CRC cell line [91]. In relation to this, it has been demonstrated that CA affects cyclin D1 expression in the same cell line. Cyclin D1 is required for G1/S transition in the cell cycle and over-expressed in many cancers. The levels of this protein are downregulated through the over-expression of the signal transducer and activator of transcription 5 (STAT5) protein and a decrease in activating transcription factor 2 (ATF-2) protein expression [92]. Overexpression of STAT5 may result in increased apoptosis, and decreased ATF-2 expression may have an anticancer action [73]. As with caffeine, a direct effect of CA has been shown on the inhibition of ERK phosphorylation, with the result of the downregulation of the neoplastic transformation of JB6 P1 cells [84]. CA also induces apoptosis and reduces invasiveness of other colorectal cell lines such as the murine CT26 cell line and in cell lines from different origin such as leukemia or endothelial cells [73]. On the contrary, Choi et al. (2015) [86] did not find any antiproliferative effect of CA or CGA on the same HT-29 cell line. However, in this case, shorter periods of incubation were assayed (20 h in [86] vs. 48–96 h in [91]).

Another important element that influences protein expression are epigenetic marks. One key factor of such regulation is the addition of methyl groups to DNA. 5-CQA and CA have resulted as strong inhibitors of DNA methylation in vitro. A DNA methyltransferase inhibition of up to 80% of normal values was achieved when the higher concentration was tested [93]. The meaning of this effect is yet to be determined.

Finally, polyphenols may also exhibit some effect on epithelial permeability. T84 CRC cells mounted in Ussing-type chambers and incubated in the presence of physiological concentrations of hydroxycinnamic acids and flavonoids showed that some of them, such as ferulic and isoferulic acids, significantly increase the expression of proteins of the tight junction complexes (zonulin 1 (ZO-1) and claudin-4) but reduce others such as occludin. In contrast, CA had no effects on the transcription of either ZO-1 or occludin [94].

#### 2.1.4. Diterpenes

Diterpenes are fatty acyl esters that have also attracted attention as coffee bioactive compounds. They are present at variable quantities in coffee beans and unfiltered coffee but in small amounts in filtered and soluble coffee [41,73]. The most studied is kahweol, which has been shown to behave as a potent inhibitor of in vitro cell viability. HT-29 CRC cells decrease their viability after exposure to kahweol at smaller concentrations than those of caffeine, CA, or CGA. This effect is mediated by an increase in the pro-apoptotic caspase-3 and a decrease in the expression of anti-apoptotic Bcl-2 and phosphorylated AKT in a dose-dependent manner [86]. The apoptotic action of kahweol has also been observed with other colorectal carcinoma cell lines (HCT116, SW480, and LoVo). In these cell lines, in addition to the HT-29 line, kahweol stimulates the activating transcription factor 3 (ATF-3), a factor known to act as a tumor suppressor in CRC that downregulates cyclin D1 and enhances p53 protein. Inhibition of ERK1/2 and glycogen synthase kinase 3 beta (GSK3β) kinases blocked kahweol-mediated ATF-3 expression [95]. Accordingly, the same authors found that kahweol decreases cyclin D1 concentration without affecting its mRNA levels. Degradation in proteasomes might be the cause of this reduction since proteasome inhibitors blocked the decrease of cyclin D1 protein levels. In accordance with this, kahweol induces the activation of ERK1/2, c-Jun N-terminal kinase (JNK), and GSK3β kinases, resulting in phosphorylation of cyclin D1, which leads to proteasomal degradation. The antiproliferative action of kahweol was not observed in the normal colon cell line CCD-18-Co [95]. In addition, kahweol may significantly attenuate the expression of the heat shock protein 70 (HSP70), causing a cytotoxic effect that is increased when cells are incubated with the chaperone inhibitor triptolide [86]. NF-kβ is another key regulatory factor implicated in inflammation and immune response that is overexpressed in many cancers [96]. Kahweol blocks NF-kβ activation through inhibiting the IkB kinase (IKK) activity. Similarly, both kahweol and cafestol, another diterpene, significantly suppress the pro-inflammatory cyclooxygenase-2 (COX-2) protein and its mRNA expression in a dose-dependent manner [97]. Anti-oxidant properties of kahweol and cafestol have also been demonstrated in non-digestive cell types such as hepatocytes, neurons, or fibroblasts, where they have been shown to be highly protective against H_2_O_2_-induced oxidative DNA damage and the generation of superoxide radicals via different mechanisms, such as the induction of cytoprotective enzymes such as the heme oxygenase-1 (HO-1) [98,99,100].

#### 2.1.5. Maillard Reaction Products: Melanoidins

Finally, melanoidins formed during the roasting process present interesting health-promoting properties. Indeed, several biological activities, such as antioxidant, antimicrobial, anticariogenic, anti-inflammatory, antihypertensive, and antiglycative activities, have been attributed to coffee melanoidins [10]. It is considered that the amount of melanoidins with antioxidant properties depend on the roasting conditions [15]. These antioxidant properties may be higher than in other sources, as shown by their ability to inhibit lipid peroxidation in an in vitro model of simulated gastric digestion [101] or in other non-digestive systems [41,102]. However, the exact mechanisms involved in such functions remain to be studied in detail.

### 2.2. In Vivo Studies

#### 2.2.1. Coffee

The potentially protective effect of coffee appeared to be supported by the first studies carried out in animals. Indeed, it was demonstrated that the chronic feeding of rodents with coffee did not increase but decreased in some cases (as in the stomach) the incidence of spontaneous tumors [103,104]. Similarly, coffee protected rats against the effects of carcinogens such as 1,2-dimethylhydrazine in the colon, although not in the small intestine [105], and it also elicited a 14-fold induction of the antioxidant and cytoprotective transferases UGT1A in the stomach of transgenic mice [77]. However, its mechanisms of action have not yet been fully elucidated. In this way, coffee consumption of over 1 cup of coffee daily in colon cancer patients has been associated with a significant attenuation of ERK, a kinase directly involved in the development of colon cancer [84]. On the other hand, differences in coffee consumers and non-consumers have been found with regard to DNA methylation levels of genes related with coffee-associated effects. The potential epigenetic action of coffee may also be mediated by sex hormones and cell type since it was only observed in women who never used hormone therapy and in mononuclear cells from blood but not from saliva [106].

Coffee has also been related with a transient damage of gastric mucosa since it increases permeability to sucrose in healthy volunteers [107].

Lastly, coffee consumption has been shown to have an impact on gut microbiota both in experimental animals and humans, even with only 3 cups a day. Decreased amounts of *Escherichia coli*, *Enterococcus* spp., *Clostridium* spp., and *Bacteroides* spp. have been reported, together with an upregulation of *Lactobacillus* spp. and *Bifidobacterium* spp. populations. In any case, the exact implications of these changes induced by coffee on the microbiota need to be determined [108,109,110].

#### 2.2.2. Caffeine

With regards to specific compounds such as caffeine, their concentration, as already mentioned, changes strongly according to coffee brand and method of preparation, which makes it very difficult to assess caffeine intake on a population basis [41].

Caffeine is rapidly absorbed in the stomach and small intestine and has been proposed to reduce cancer risk by altering the metabolism of carcinogens such as 2-amino-1-methyl-6-phenylimidazo (4,5-*b*) pyridine (PhIP), as shown in rats. PhIP is an amine to which humans are strongly exposed from cooked meat and fish and, accordingly, it has been implicated in CRC. Regarding this, coffee has been shown to increase the expression of enzymes involved in the detoxification of PhIP, such as glutathione *S*-transferase (GST) [111]. As a result, caffeine decreased the number of PhIP-induced colonic aberrant crypt foci (ACF) preneoplastic lesions [112]. Interestingly, a study with human healthy volunteers showed that unfiltered coffee elicited an increase in the detoxification capability and anti-mutagenic properties of the colorectal mucosa through an increase in glutathione concentration [113]. However, it is important to note that when the carcinogenic effect of PhIP was combined with a fatty diet, cell proliferation increased without caffeine being able to prevent it, which is a factor to be considered when interpreting epidemiological studies [114]. Likewise, in a rat CRC model induced by N-methyl-N-nitro-N-nitrosoguanidine (MNNG), coffee decreased the development of dysplastic crypts in a caffeine-independent way, although both decaffeinated coffee and caffeine decreased inflammatory stress and DNA damage [115]. On the contrary, previous studies with a model of stomach carcinogenesis induced in rats by MNNG and NaCl showed that lipid peroxidation in the glandular stomach mucosa was inhibited by caffeine treatment, resulting in less gastric tumors [116]. Differences in caffeine dosage and administration and method of tumor induction might explain these contradictory results.

#### 2.2.3. Polyphenols

Regarding polyphenols, studying ileostomy subjects, Olthof et al. (2001) [117] determined that about one-third of CGA is absorbed in the small intestine. The rest of the polyphenols reach the colon, where simpler molecules are produced by breakdown due to microbial activity and, thus, very few of the absorbed molecules retain the structure of the parent CQAs present in coffee. Therefore, microbial action is necessary for absorption of phenols but also the individual microbiome itself is modulated by them [70,72,118]. As a result, the CQA derivatives that are ultimately absorbed are very varied. It is not clear then whether they prevent or induce cell damage, and their overall effect in the body is far from being understood since the number of studies regarding CGA effects in vivo is very limited [41,73]. In any case, CA might be implicated in the decrease of cancer metabolism. Kang et al. (2011) [84] showed that lung metastasis induced in mice by infusion of CT-26 colon cancer cells are inhibited after CA administration. CA strongly suppresses the activity of mitogen-activated MAPK/ERK kinase (MEK1), a protein kinase whose constitutive activation results in cell transformation, and TOPK, a serine/threonine kinase expressed in high levels in CRC and an activator of ERKs. CA binds directly to either MEK1 or lymphokine-activated killer t-cell-originated protein kinase-like protein (TOPK) in an ATP-noncompetitive manner.

#### 2.2.4. Diterpenes

A third group of bioactive compounds assayed in vivo are the diterpenes cafestol and kahweol. These compounds act in rats as chemoprotectants against PhIP. In this case, PhIP–DNA adduct formation in the colon was reduced up to 54% compared to controls. Similarly, these diterpenes reduced buccal carcinogenesis in hamster after dimethylbenz (a) anthracene (DMBA) treatment [119]. The effects of kahweol and cafestol seem dependent on the continuous presence of these compounds in the diet since they are reversible following their removal. These detoxicant effects might be mediated by the ability of kahweol and cafestol to induce GST and other metabolic enzymes such as UGT1A. In this way, it has been shown that diterpene ingestion results in 2.5-fold induction of GST and a dose-dependent increase in UGT1A in rats [99,120]. In mice, kahweol seems to be a more potent inductor of GST than cafestol [121]. A chemoprotective role of kahweol and cafestol by the direct prevention of carcinogen–DNA binding should also not be discarded [99].

The studies on the effects of these diterpenes in humans are scarce but an increase in total cholesterol in serum has been reported [122]. However, the variability in the concentration of cafestol and kahweol in commercial coffee makes it difficult to answer the question as to whether the beneficial effects observed in animals may occur in humans consuming moderate amounts of coffee without inducing hypercholesterolemia. This question remains open, although considering the difference between the dose needed for hypercholesterolemia and for enzyme induction in model animals, beneficial effects in humans might also be expected without an increase in blood cholesterol [99].

#### 2.2.5. Maillard Reaction Products: Melanoidins and Acrylamide

Maillard reaction is a main chemical event taking place during coffee roasting. Melanoidins are produced during coffee roasting due to the Maillard reaction. As mentioned above, melanoidins display a wide range of beneficial properties. In a mouse model of IBD induced by dextran sodium sulfate (DSS), a correlation between the exposure to melanoidins and the attenuation of inflammation has been shown, although the precise mechanisms involved remain to be elucidated [123]. It would be interesting to investigate if their antioxidant and metal chelating activity, antimicrobial activity, and the ability to modulate colonic microflora, as well as antihypertensive activities described in vitro (see above), are reproducible in model animals.

Finally, it is worth mentioning that melanoidins behave in vivo as dietary fiber, being largely indigestible by humans and fermented in the gut [41,102]. Melanoidins might be relevant contributors to colonic health since their intake may reach up to 20% of the recommended daily intake of dietary fiber [15]. This will be more deeply discussed in the following section, focused on the effects of coffee and its components on gastrointestinal motility.

Acrylamide is also produced during roasting of coffee due to the high temperatures employed in this step of the food processing. After absorption, a significant fraction of acrylamide is converted metabolically to the chemically reactive and genotoxic glycidamide. Acrylamide is a very soluble carcinogen that has been shown to cause tumors in experimental animals at multiple organ sites, but not in digestive organs [124]. Interestingly, epidemiological studies have failed thus far to provide evidence of an increased risk of most types of cancer after exposure to acrylamide in humans [124,125].

However, acrylamide is not without effects on the gastrointestinal epithelium, as some reports in experimental animals (rats) have shown vascular congestion, mucosal erosions, and depletion of the protective surface mucus together with widespread inflammatory infiltration in gastric samples after 4-week oral administration of acrylamide at 30 mg/kg, due to severe oxidative stress manifested as a significant increase in lipid peroxidation and depletion of antioxidant enzymes in gastric tissue, as well as, likely, high nitric oxide (NO) production after inducible nitric oxide synthase (iNOS) induction [126].

## 3. Coffee and the Gastrointestinal Tract: Focus on Motor Function

Gastrointestinal motility is a complex process involving different elements. The element more directly responsible of motor function is the smooth muscle, of which two layers are found in all gastrointestinal organs: the circular (inner and thicker) and the longitudinal (outer and thinner) layers (their names refer to the orientation of their smooth muscle cells, around or along the longitudinal axis of the gastrointestinal tract; the stomach has an additional, oblique smooth muscle layer).

Between the inner and the outer muscle layers lie the myenteric plexus, the part of the enteric nervous system (ENS, intrinsic innervation of the gastrointestinal tract) directly responsible of gastrointestinal motor function [127]. In the myenteric plexus, different subpopulations of myenteric neurons participate in the generation of the different motor patterns, such as the peristaltic reflex, i.e., the basic motor pattern that allows the luminal contents to progress distally thanks to the oral contraction and aboral relaxation of the circular muscle, together with coordinated changes in the length of the longitudinal muscle. The enteric glial cells (which were previously considered simply as supportive cells for neurons, but now are recognized to exert important signaling functions) also collaborate to coordinate motility [128]. In addition, the interstitial cells of Cajal (ICC), located at different levels within the muscle layers and myenteric plexus, play a pacemaker role and generate stereotyped activity patterns (i.e., slow waves [129]).

Moreover, extrinsic innervation from the autonomic nervous system (vagal and pelvic nerves belonging to its parasympathetic branch, and splanchnic nerves belonging to the sympathetic branch) as well as hormones secreted within the gut wall (from enterochromaffin cells (ECs), L cells, etc.) or reaching the gut wall via the blood stream from different extrinsic endocrine glands, are classically recognized as important modulators of gastrointestinal tract motor function.

Finally, immune cells (mast cells in particular [127,130]) and microbiota may produce and release mediators and metabolites, respectively, that may remarkably alter motility and either contribute to maintain a healthy gut or facilitate the development of gut disorders.

The effects of coffee and its components on gastrointestinal tract motor function in general and the specific mechanisms involved have been relatively scarcely evaluated.

### 3.1. Effects of Coffee Brew on Gastrointestinal Motility

Despite the popular use of coffee around the world, the effects of this beverage on gastrointestinal motor function have surprisingly been evaluated only scarcely, especially when compared with those on other systems such as the cardiovascular and the central nervous systems. It was soon demonstrated that coffee reduces lower esophageal sphincter pressure [131] and stimulates secretion from the stomach [69]. Both effects may cause or aggravate heartburn, the most frequent effect attributed to coffee. This might be caused by direct irritation of the esophageal mucosa or by promoting GERD [132], which may favor the development of Barrett’s esophagus (BE) (see above).

Using a barostat, coffee was found to prolong the adaptive relaxation of the proximal stomach, compared with an isotonic control solution, suggesting that it might slow gastric emptying [132]. However, other studies, using scintigraphy or applied potential tomography, indicate a lack of effect in stomach motor function, or even accelerated gastric emptying, in a portion of individuals [132,133,134,135]. These conflicting results may have been due to methodological differences, including the selection of participants (healthy or dyspeptic) or the type of coffee drink used for the studies [132]. Despite early associations of coffee drinking and functional dyspepsia [136], these were later attributed to study bias related with characteristics of patients (higher adiposity [137], more attentive to their symptoms [138]). Indeed, despite the relaxing effects of the proximal stomach, coffee did not modify gastric wall compliance, wall tension, or sensory function [139]. No association has been confirmed to occur either with peptic ulcer disease [132].

Caffeine-containing beverages (75–300 mg) were shown to induce a dose-related secretion from the small intestine [140], although coffee by itself had no significant effect on sodium and water transport, maybe due to compensatory effects by other coffee components [141]. Despite the common believe that coffee favors diarrhea, the effects on jejunal and ileal fluid secretion were not associated with changes in small bowel transit [140]. Orocecal transit studies did not find any significant effect of coffee compared with control solutions either [135]. However, these results could be due to the use of lactulose as a substrate for assessing transit, since this compound by itself accelerates transit and may have masked the possible effects of coffee [132,142].

In an early study, using radiographic techniques, it was shown that coffee, drunk together with a low-fat breakfast, induced a contraction of the gallbladder that was similar for regular and caffeinated coffee [143]. Years after, this was also demonstrated for regular coffee using ultrasonography, although no control drink was used as a comparator [144]. Further research, using a better controlled experimental design, confirmed that caffeinated and decaffeinated coffee induces cholecystokinin release and gallbladder contraction [145], which may explain why patients with symptomatic gallstones often avoid drinking coffee.

Regarding colonic motility, it was soon found that coffee, either caffeinated or not, promotes the desire to defecate at least in one-third of the population, predominantly women, and that this was associated with an increase in rectosigmoid motor activity. Furthermore, it was found that this increase occurred as soon as 4 min after drinking the coffee, caffeinated or not, but not after drinking hot water. Since (unsweetened) coffee contains no calories, and its effects on the gastrointestinal tract cannot be justified by its volume load, acidity, or osmolality, it was soon recognized that it must have pharmacological effects [132]. Thus, these findings were interpreted as mediated indirectly by a component of coffee other than caffeine, which, by acting on epithelial receptors in the stomach or small bowel, would trigger a gastrocolonic response, speculated at that time to be due to the release of cholecystokinin or another hormone [146].

Interestingly, these results were further supported using ambulatory colonic manometry [147]. In a study with 12 healthy volunteers, a probe was placed up to the mid-transverse colon and the following day the effects of four different drinks were evaluated: unsweetened black coffee, unsweetened decaffeinated coffee, a 1000 kcal meal, and water. Caffeinated coffee significantly increased colonic motor activity, including both propagated and simultaneous contractions, which were 60% and 23% greater than those of water and decaffeinated coffee, respectively, and similar to the effect of the meal. Both the caffeinated coffee and the meal (but not the decaffeinated coffee) produced a strong gastrocolonic response, but no significant effect of gender was detected in this case. Compared to water, coffee increased propagating contractions by 50%, suggesting that this drink may induce propulsive motor activity, and this was accompanied by a higher incidence of abdominal cramps, flatulence, and urination, leading to confirmation of the popular belief that coffee stimulates colonic motor activity. The effect of caffeinated coffee was similar to that of the meal during the first 30 min, although it was of shorter duration (1–1.5 h vs. 2–2.5 h). Decaffeinated coffee seemed to also enhance colonic motor activity but was less potent than caffeinated coffee and seemed to exert this effect only at the more proximal colonic sites recorded. The short latency of the effect of coffee (of either kind) after its ingestion was again interpreted as due to the involvement of an indirect mechanism, probably a neurohumoral response mediated by the small bowel, since gastric emptying of coffee occurs as soon as 15–20 min [148]. The authors acknowledged that the specific molecule involved was not yet known but mentioned different possibilities, such as cholecystokinin, exorphins (opioid-like molecules present in coffee), gastrin, or motilin, as well as the fact that other active ingredients contained in coffee could add their own direct effects on gut smooth muscle. Importantly, the authors suggested that the effect demonstrated for coffee could be beneficial for patients with colonic disorders such as slow transit constipation but might be detrimental to patients with diarrhea or fecal incontinence [147].

In these regards, Gkegkes et al. recently published a systematic review and meta-analysis in which they evaluated the evidence suggesting a potential role of coffee to prevent postoperative ileus [149]. Postoperative ileus is a significant complication of surgery, and management is not yet optimal. The underlying cause of this clinically relevant problem is multiple, including surgical manipulation itself, opioid analgesics, inflammation, electrolyte fluctuations, and imbalances in the autonomic function and gastrointestinal hormonal system [150,151]. Although frequently self-resolving, postoperative ileus represents an important clinical and economic burden, particularly hospital expenses, due to delayed discharge [152]. In addition to other measures, prokinetic agents (alvimopam, ghrelin agonists, neostigmine and serotonin receptor antagonists), chewing gum, gastrograffin, and coffee are used for management. The authors focused on coffee and found four randomized controlled trials as eligible for their study, with three of them referring to colorectal procedures [153,154,155] and only one to gynecological surgery [150], with a total of 341 patients (the sample size in each study was 58–114 patients). Coffee was administered postoperatively to 156 patients. The most remarkable results were (1) coffee did not significantly increase complications compared with the control group; (2) coffee significantly decreased the time period until the first bowel movement, as well as the time to tolerance of solid food, the first flatus, and the first defecation; (3) no significant effects were found regarding the length of hospital stay. Decaffeinated coffee was proved to reduce time to initiation of bowel movement [155], suggesting that caffeine is not necessary for coffee effect, and it was suggested that maybe CGAs and melanoidins could have a role [15]. Both display antioxidant effects, whereas melanoidins may contribute a fiber effect to coffee anti-ileus properties (see below). Moreover, the authors proposed that other chemically active agents might be formed during decaffeination [149]. Other mechanisms that may contribute to the positive effects of coffee are related with the anti-inflammatory effects of some of its compounds. In these regards, C-reactive protein (CRP) levels, used as a marker of postoperative complications, were found to be significantly lower in patients given coffee after removal of the nasogastric tube on the first postoperative day compared with the group of patients that were not given coffee. Furthermore, in that study, lower CRP levels were associated with reduced time to initiation of bowel movements, as well as reduced rates of postsurgical complications and hospitalization time, particularly in patients with a right colon tumor [155]. Another important conclusion of this meta-analysis is that, compared with alvimopam (a peripheral opioid antagonist), which also showed good results to reduce the constipating impact of opioid use associated with surgery [156,157], coffee may be a much cheaper therapeutic strategy to achieve comparable results [149]. As in other studies using coffee, the limitations highlighted by the authors include the differences in quality and quantity of administered coffee, together with low number of participants and heterogeneity of patients and operations.

In general, it can be said that studies performed thus far evaluating the effects of coffee in humans are relatively scarce, with relatively low numbers of participants and mainly healthy (other than those participating in the studies related with postoperative ileus), with high heterogeneity in the kind of coffee used and not too high methodological quality. In addition, no animal study could be found using coffee itself to test motility-related parameters, in contrast with studies of the effects of particular coffee components, as will be discussed next.

### 3.2. Caffeine

In vitro studies have mainly tested the effects of caffeine, whose pharmacology is quite complex. Thus, caffeine is a non-selective adenosinergic antagonist. In addition, in many cell types, caffeine releases calcium (Ca^2+^) from internal stores through ryanodine receptors (RyR) and also increases the content of cyclic adenosine monophosphate (cAMP) by inhibiting the activity of phosphodiesterases [158].

Interestingly, caffeine has been used as a tool to investigate the contractile and/or electrical properties of the different components of the gut wall involved in motor function along the gastrointestinal tract [159], including the myenteric plexus (both neurons and glial cells), the smooth muscle cells, and ICCs, as well as their dependency on intracellular calcium dynamics.

The effects exerted in vitro by caffeine in the gastrointestinal smooth muscle have been tested using different techniques, including the recording of contractile activity of smooth muscle strips (which contain also myenteric plexus and ICCs) in organ baths, and the electrophysiological recording of cultured single smooth muscle cells. In these experiments, the effects of caffeine were shown to be dose-dependent, with low doses (0.1–0.3 mM) relaxing and high doses (>0.3 mM) producing a transient contraction followed by relaxation [160].

Moreover, caffeine at relatively high doses (1–10 mM) inhibits slow waves (generated by ICCs) in different gastrointestinal tissues from different species [161], including the human jejunum [162]. In addition, despite early reports of no effects of caffeine on glial cells [163], more recent studies have shown that caffeine at 0.01 mM produced an immediate and sustained Ca^2+^ response in all myenteric glial cells from mouse colon, confirming that they have ryanodine-sensitive Ca^2+^ stores [164]. Thus, caffeine might modulate glial cell function at relatively low doses, and this, in turn, may have an impact on gastrointestinal motor activity through coordinated responses with the myenteric neurons.

The effect of caffeine on the myenteric neurons has been studied using cultured isolated neurons/ganglia or whole-mount preparations (see Figure 1 and Figure 2). In cultured myenteric neurons, caffeine was shown to concentration-dependently stimulate Ca^2+^ release, in a quantal and saturable manner, from intracellular Ca^2+^ stores that are refilled via depolarization-induced Ca^2+^ increases. This effect was shown to be sensitive to the RyR antagonists ryanodine, dantrolene, and procaine, but did not involve the participation of cAMP phosphodiesterase inhibition [163]. However, caffeine-releasable ryanodine-sensitive calcium stores are not the only subset of cytosolic Ca^2+^ storage and the calcium ionophore ionomycin applied after maximal caffeine effect in Fura-2-loaded myenteric neurons achieved further increases in intracellular free Ca^2+^ ([Ca^2+^]i) [163]. An important drawback of these studies in cultured myenteric neurons/ganglia is that it is difficult to identify the functional subpopulations of myenteric neurons, although heterogeneity of the responses to caffeine (as well as to other drugs) is clearly observed, suggesting that it might affect different neuronal subtypes [163].

In order to evaluate the effects of drugs on specific subpopulations of myenteric neurons, the use of whole-mount preparations is a better option. Whole-mount preparations are “sheets” of longitudinal muscle with the myenteric plexus attached. The other gut wall layers, i.e., mucosa, submucosa, and circular muscle, are dissected away to facilitate intracellular recording of myenteric neuron electrical activity. In addition, these experiments allow for the definition of the morphology of the impaled neuron through intracellular injection of a marker during recording, as well as its chemical code, by use of immunohistochemistry after fixation (Figure 2).

In these experiments, ryanodin-sensitive calcium stores were demonstrated to play a particularly important role in a specific subpopulation of myenteric neurons whose electrophysiological features (in whole-mount preparations) highly depend on [Ca^2+^]i, the so-called AH/type II neurons (Figure 3), which are identified as intrinsic primary afferent neurons [166]. Morphologically, these are multipolar neurons with a smooth soma and projections to the mucosa and other myenteric neurons. Electrophysiologically, these neurons are characterized by relatively broad action potentials (APs) (i.e., the falling phase of their APs display a “hump”), followed by two afterhyperpolarizations (AHPs); an early, short (ms) AHP; and a late, long-lasting (4–20 s) AHP [166]. Similar to cardiomyocytes, the broad AP is due to the influx of both sodium and calcium through voltage-gated channels. Importantly, Ca^2+^ entry during the AP is associated with a transient increase in [Ca^2+^]i, released from RyR-sensitive stores, which amplifies calcium influx. This calcium-induced calcium release (CICR), in turn, leads to potassium efflux through calcium-operated potassium channels, which underlies the characteristic slow AHP of these neurons. Thus, activity-dependent CICR has been suggested to be a mechanism to grade the output of AH neurons according to the intensity of sensory input. Furthermore, AH neurons display relatively high resting levels of [Ca^2+^]i, which, through the same indirect mechanism involving potassium efflux, maintain low their resting potential and reduce their excitability [167].

Interestingly, activation of AH myenteric neurons by caffeine turns into a reduction of their excitability due to the increase in [Ca^2+^]i released from ryanodine-dependent stores and the consequent potassium-mediated hyperpolarization [167,168]. How this translates into in vivo effects is not clear, but it is important to remember that AH/type II myenteric neurons extend projections to the mucosa where caffeine might activate them directly, leading to the indirect inhibitory effect suggested by these in vitro studies.

### 3.3. Polyphenols

Wood and collaborators used intracellular recording of enteric neurons and CA as a tool to understand the mechanisms underlying pathophysiology of secretory diarrhea associated with food allergies [169]. Antigen-evoked mast cell degranulation in the small and large intestine starts an immediate (type I) hypersensitivity reaction characterized by mucosal hypersecretion [170] and strong contractions of the musculature [171], which are sensitive to tetrodotoxin and atropine, meaning that these reactions implicate the participation of the ENS. Exposure to an antigen, then, triggers a coordinated immunoneural defense program response aimed at getting rid of the antigenic threat, which translates into watery diarrhea, fecal urgency, and abdominal pain [172]. In the study by Wood et al., CA, which is a 5-lipoxygenase inhibitor [173], was able to partially suppress the hyperexcitability of submucous neurons induced by β-lactoglobulin in the small intestine of guinea pigs, which was used as a model of anaphylactic responses associated with food allergies [169]. Thus, this study demonstrated the involvement of leukotrienes in the secretory response of submucous neurons to food antigens. However, to our knowledge, the in vitro effects of CA on the myenteric neurons in the context of food allergy has not been tested thus far.

Of note, CA and CGA may exert an important neuroprotective role in the context of Parkinson’s disease (PD), including on the ENS, which is discussed in Section 4. However, to our knowledge, the potential impact of these polyphenols on gastrointestinal motility altered in preclinical models of PD or in patients, has not been specifically evaluated.

### 3.4. Dietary Fiber

Dietary fiber is also present in coffee. Our group evaluated the effect of two coffee-derived by-products proposed as natural sources of dietary fiber on in vivo rat gastrointestinal motility using radiographic methods after intragastric administration of barium as contrast (Figure 1B illustrates this procedure). In one of them, different experiments were performed to evaluate the properties of instant spent coffee grounds (SCGs) [174]. The gastrointestinal motility study was performed after the 1st, 14th, and 28th intragastric administration of SCGs. The product accelerated transit in both the small intestine (since the cecum of the rats was reached significantly more quickly than that of control rats), and the colon (since formation of fecal pellets occurred also much more quickly in SCGs-exposed than in vehicle-treated animals). However, this effect was restricted to the first radiographic session, after the first SCG dose, whereas it was not apparent after the 14th or the 28th doses. Thus, the dietary fiber effect acutely produced by SCG administration seemed to be followed by tolerance development, although no sign of impaired motility was found in any animal. The acute pro-motility effects of SCG demonstrated in that study could be influenced by short chain fatty acids (SCFAs), which have been shown to be released during SCGs fermentation by colon microbiota from medium and dark roasted coffee beans at concentrations higher than 10 mM [175], and SCFAs (10–200 mM) stimulate colonic motility [176]. Interestingly, an aqueous extract of coffee silverskin, another coffee by-product, may also display dietary fiber effect since total SCFAs derived from coffee silverskin extract fermentation were higher in feces of rats treated with the extract for 28 days [177]. Although its precise in vivo effects on gastrointestinal motility still needs to be determined, it might be related with the presence of melanoidins in it (see Section 3.5).

### 3.5. Maillard Reaction Products: Melanoidins and Acrylamide

The other radiographic study of gastrointestinal motility performed by our group evaluated the effect of melanoidins from the previously mentioned aqueous extract of coffee silverskin [178]. Coffee silverskin is the tegument of the outer layer of the coffee bean, representing approximately 4.2% (w/w) of the coffee cherry and is the only by-product produced during coffee roasting [179]. Coffee silverskin has been proposed as a sustainable natural source of prebiotics, antioxidants, and dietary fiber [180]. The antioxidant properties of coffee silverskin extract are due to the presence of CGA [181], but also to the melanoidins generated during the roasting process [182]. Melanoidins are high molecular weight brown polymeric compounds generated during the last stage of the Maillard reaction [183], and those derived from coffee are described as “Maillardized dietary fiber” [184]. Thus, the fiber effect was studied in vivo, in healthy male Wistar rats, at a dose of 1 g/kg in drinking water. After 4 weeks, the rats received barium sulphate by gavage and radiographs were taken 0–8 h afterwards. In addition, the colonic bead expulsion test was performed to specifically determine the possible effects on colonic propulsion. In line with the previous study with SCGs, melanoidins accelerated transit in small intestine (since the caecum was reached significantly more quickly in melanoidin-exposed rats than in control animals) and tended to accelerate fecal pellet formation, although this effect was not significant. Interestingly, fecal pellets from the melanoidin group tended to be slightly larger, which might have been a result of the higher fiber intake in this group, making the fecal pellets slightly more effective to mechanically stimulate the colon. Moreover, melanoidins did not significantly alter the latency of expulsion of a bead inserted 3 cm into the colorectum, suggesting that the motor agents (intrinsic and extrinsic innervation, smooth muscle, and ICCs) involved in the colonic propulsion at this level were not altered by dietary exposure to coffee silverskin-derived melanoidins and that these may have potential to be used as a functional food ingredient [178]. Interestingly, Argirova et al. (2010) [185] showed that melanoidins act upon muscle tone and might facilitate Ca^2+^ influx into the cells of isolated gastric muscle layers. Thus, these compounds might exert their pro-motility action not only through a fiber effect, but also through the direct activation of gastrointestinal smooth muscle cells, which needs to be confirmed to occur also using isolated intestinal muscle tissue.

As mentioned above, acrylamide is formed as a result of Maillard reaction between amino acids and sugars during heating [186], which occurs also during coffee roasting. Although the dietary intake of acrylamide in humans is difficult to assess, the estimated dietary exposure for the general population ranges from 0.3 to 0.8 μg/kg of body weight per day [187]. This is due to the exposure, not only to coffee, but also to other foods (chips, cereals) and industrial products (those related with polymer, glues and paper, water treatment, and cosmetic industries [126]) that may also contain acrylamide in relevant concentrations to impact human health. Despite the fact that the digestive tract is one of the main routes of acrylamide absorption, and the fact that intake of acrylamide-containing foods, including coffee, is still growing, its effect on the ENS neurons has scarcely been assessed, yet this is important because acrylamide is a peripheral nervous system toxin.

Acrylamide effects were studied in a coculture model of intestinal myenteric neurons, smooth muscle cells, and glia [188]. In this study, acrylamide was added to the cocultures in doses ranging from 0.01 mM to 12 mM, followed by incubation for 24, 96, or 144 h. In contrast with botulinum toxin A, which was also tested in the same system and only altered neuronal function, acrylamide damaged enteric neuron structure when used at 0.5–2 mM. At these doses, the damage was selective for the axonal structures, without affecting survival, whereas at higher doses, neuronal survival was significantly reduced. Axonal loss was accompanied by reduced acetylcholine release, which was negligible at 4 mM. The mechanism involved synaptic vesicle synthesis and function, but not choline uptake. Neuronal loss at high doses involved mainly a necrotic mechanism, although a low frequency of noncaspase-3-mediated apoptotic death was also demonstrated. Interestingly, it was also shown that after low-dose acrylamide challenge, axon regeneration was possible. In fact, axon growth occurred more rapidly than in control cultures over the 24–96 h period after low-dose challenge, suggesting the involvement of compensatory mechanisms after the initial damaging insult. However, neurotransmitter release was found to lag behind axonal regrowth by at least several days. Interestingly, all the changes described were selective to neurons (irrespective of the underlying phenotype), with enteric glial cells apparently not being affected [188].

Acrylamide has also been shown to produce neurotoxic effects on the ENS after oral administration to experimental animals. Early studies showed changes in the ENS in acrylamide-treated rats resembling those of streptozotocin-induced diabetic animals, with alterations of the catecholaminergic content, a decrease in the amount of calcitonin gene-related peptide (CGRP), and a corresponding increase in the levels of vasoactive intestinal peptide (VIP) [189]. However, those studies did not evaluate whether these changes were associated with neuronal loss, axonal degeneration, or altered function. More recently, the effects of acrylamide administration have been studied in pig models. The results suggest that even low doses of acrylamide affect the structure and function of the gastrointestinal tract and cause a significant response of ENS neurons. For example, the expression of cocaine- and amphetamine-regulated transcript (CART), which exerts a crucial role in neuronal response to stress stimuli and neuroprotection, was increased, particularly in the myenteric plexus of the small intestine in immature female pigs receiving a low or a high dose of acrylamide by the oral route for 28 days, which was interpreted to occur as part of the neuronal protection/recovery processes within the gastrointestinal tract in response to this pathological stimulus [190]. Galanin is another peptide with neuroprotective properties that mediates survival or regeneration after neural injury and exerts anti-inflammatory activities [191,192]. Thus, in the same porcine model, the population of galanin-like immunoreactive neurons was found to be increased in both the submucous and myenteric plexuses of the stomach, even at low doses. Moreover, the submucous and myenteric neuronal populations of cells immunoreactive to galanin and simultaneously immunoreactive to VIP, nNOS, or CART were increased. These findings were, again, interpreted by the authors as a neurotrophic/neuroprotective role of galanin (in possible co-operation with VIP, nNOS, and CART) in the recovery processes in the gastric ENS after acrylamide intoxication [193]. One more paper of this series was published in the year 2019 and found similar results in the pig duodenum. As before, acrylamide was used at a low (0.5 μg/kg) daily dose considered tolerable, or at a 10-fold dose (5 μg/kg), by the oral route for 4 weeks. Both treatments led to significant increases in the percentage of neurons immunoreactive to substance P (SP), CGRP, galanin, nNOS, and vesicular acetylcholine transporter (VACHT), although the high dose exerted more intense changes. In this case too, the interpretation given by the authors is that all these changes may be compensatory plastic effects in an attempt to protect neurons from damage and restore enteric neuronal homeostasis [194]. Of note, although acrylamide activates microglial cells both in vivo and in vitro, leading to the release of proinflammatory cytokines, and consequently contributing to neuronal damage [195], the involvement of enteric glial cells in enteric neuronal alterations induced by acrylamide has not been specifically evaluated yet, except for the above-mentioned study that used cocultures of intestinal myenteric neurons, smooth muscle cells, and glia, and did not show any acrylamide-induced alterations in this last cell type [188].

## 4. Coffee and the Brain–Gut Axis

As already mentioned, coffee is a natural source of compounds (Figure 4) able to exert crucial effects in the brain–gut axis [196].

Interestingly, when “brain–gut axis” and “coffee” are combined as keywords in Pubmed, only three papers are retrieved (as of 29th November 2020), and two of them lead with the relationship of coffee with PD (see below) [197,198]. The other one is a recent study by Papakonstantinou et al. [199], who performed a randomized, double blind, crossover clinical trial (ClinicalTrials.gov ID: NCT02253628) with 40 healthy young (20–55 years of age) individuals of both sexes to study the effect of 200 mL coffee beverages containing 160 mg caffeine (hot and cold instant coffee, cold espresso, hot filtered coffee) on (1) self-reported gastrointestinal symptoms, (2) salivary gastrin, (3) stress indices (salivary cortisol and α-amylase) and psychometric measures, and (4) blood pressure. Importantly, the participants were daily coffee consumers, and the study was performed in non-stressful conditions. There was no effect of coffee on self-reported anxiety levels. Furthermore, the participants reported a very low score (1 out of 10) for all the questions pertaining to negative gastrointestinal symptoms (i.e., abdominal discomfort, bloating, dyspepsia, and heartburn), chronic stress, and negative feelings, whereas the score was high (9 out of 10) for all the questions pertaining to positive feelings. Coffee consumption significantly increased salivary α-amylase activity, with significant differences only between cold instant and filtered coffee at 15 and 30 min after intake. Irrespective of coffee type, salivary gastrin was temporarily increased, whereas salivary cortisol or self-reported anxiety levels were not affected. However, at the end of the experimental periods, blood pressure was significantly increased (but within the healthy physiological levels), independently of coffee type/temperature. Although many studies have addressed the cardiovascular and central effects of coffee and caffeine, the report by Papakonstantinou et al. seems to be the only study specifically evaluating their effects on the brain–gut axis as a whole, in the same individuals and under the same conditions. Thus, it was demonstrated that acute coffee consumption in non-stressful conditions was not associated with gastrointestinal symptoms but activated the sympathetic nervous system, associated with increases in salivary α-amylase and blood pressure but not salivary cortisol, which was interpreted as due to a possible anti-stress effect of coffee [199], possibly contributed by a coffee compound other than caffeine. Thus, it is important to study the effects not only of coffee, but also of its components, on the brain–gut axis.

### 4.1. Caffeine

Caffeine is the main psychoactive compound found in coffee (Table 1). It is consumed from the diet and absorbed into the blood, stimulates the sympathetic nervous system activity, and easily crosses the blood–brain barrier (BBB) with stimulatory effects also on the central nervous system (CNS) [196,200]. Caffeine has an effect on the CNS by modulating different neuronal pathways.

Thus, both in animal and human studies, changes in dopaminergic systems have been observed after caffeine exposure [201]. Different studies suggest that caffeine increases extracellular dopamine concentrations [202], as well as the expression of dopaminergic receptors [203] and transporters, which leads to an improvement of cognitive dysfunction and attention [204]. In addition, it has been reported that caffeine is able to combat the loss of dopaminergic neurons, inducing neuroprotection and attenuating neurological diseases in animal models [205], which may be particularly useful in the context of PD (see below). However, dopaminergic activity is increased in schizophrenia and addictions. Therefore, the effects of coffee and caffeine must be considered also in these patients. Importantly, people with schizophrenia have relatively high intakes of coffee and caffeine, due to different reasons, including the willingness to relieve boredom and apathy or the side effects of antipsychotic medication, such as sedation or dry mouth [206]. In general, it is recommended that these patients reduce coffee consumption [207].

On the other hand, it has been reported that there is a possible interaction between caffeine and glutamatergic signaling. Chronic caffeine consumption can attenuate blast-induced memory deficit in adult male C57BL/6 mice, which is correlated with neuroprotective effects against glutamate excitotoxicity, inflammation, astrogliosis, and neuronal loss at different stages of injury [208]. Moreover, caffeine consumption could also reduce the loss of glutamatergic nerve terminals in the hippocampus, restoring diabetes-induced memory dysfunction in mice [209].

Additionally, caffeine has been found to decrease the activity of the γ-aminobutyric acid (GABA) ergic system and modulate GABA receptors, leading to neurobehavioral effects [201]. Chronic caffeine intake may be related to long-term decrease in GABA [210].

Finally, a recent review by Jee et al. (2020) has indicated that caffeine consumption has different neurological and psychiatric effects in men and women [211], highlighting the importance of evaluating the influence of gender on the effects of coffee and its components on the brain–gut axis. Specifically, the authors showed that caffeine consumption reduces the risk of stroke, dementia, and depression in women and of PD in men. However, caffeine has a negative effect of increasing sleep disorders and anxiety in both male and female adolescents [211].

### 4.2. Polyphenols

Coffee is also a source of CGA (Table 1), a hydroxycinnamic acid with health-promoting properties such as antioxidant, antibacterial, and anti-inflammatory activities, among others [212]. The majority of ingested CGA is hydrolyzed to CA and quinic acid, and further metabolized by gut microbiota into various aromatic acid metabolites [213]. There is controversial information regarding the ability of CGA and its metabolites to cross the BBB [214,215]. However, neuroprotective effects due to its antioxidant and anti-inflammatory properties have been previously described [215].

As mentioned for caffeine, CA and CGA are coffee components with antioxidant properties and neuroprotective effects against dopaminergic neurotoxicity [216,217] that have been suggested to underlie the decrease in PD risk associated with coffee consumption [218,219]. Interestingly, one of the cardinal symptoms of PD is constipation, and this seems to occur already 10–20 years prior to the presentation of PD motor symptoms [220], with lower bowel movement frequencies predicting the future PD crisis [221]. Moreover, neurodegeneration occurs in PD patients and animal models, and robust evidence suggests that PD could start in the ENS and spread from there to the CNS via the vagal nerve [222,223]. In a recent report, CA or CGA were tested in a rotenone-induced PD mouse model [224]. In this model, mice were subcutaneously implanted an osmotic mini pump, allowing the administration of rotenone at 2.5 mg/kg/day (corresponding to environmental exposure levels of rotenone via pesticides) for 4 weeks. CA (30 mg/kg/day) or CGA (50 g/kg/day) were administered 5 days a week, starting 1 week before rotenone exposure, up to its end. The effects of treatments on central dopaminergic and enteric neurons were evaluated after sacrifice and performed 1 day after rotenone treatment ended. In addition, cultures of rat enteric neurons and glial cells were exposed to rotenone (1–5 nM) with or without exposure to CA (10 or 25 μM) or CGA (25 μM). Remarkably, besides beneficial effects on central structures and cells relevant to PD (namely, nigral dopaminergic neurons and astrocytes), it was demonstrated that administration of CA or CGA at least partially prevented the changes induced by rotenone, which affected both neurons and enteric glial cells in the intestinal myenteric plexus of treated mice. Importantly, all these effects were reproduced in vitro. The precise mechanism was not clarified but it was proposed that CA or CGA pretreatment may enhance the reactivity of glial cells to produce antioxidative molecules in response to rotenone exposure. Although the CA and CGA doses used were probably 2–5 times higher than those consumed daily by coffee drinkers, the results are clearly promising. In fact, the authors suggested that it may be possible to use a food-based promising therapeutic strategy of neuroprotection to improve not only motor but also non-motor symptoms of PD, such as constipation, although the effects of CA and CGA on gastrointestinal motility were not specifically evaluated in this report [224].

During the last stage of preparation of this manuscript, a report by the group of Rogulja [225] was published showing that there is a key connection between beneficial effects of sleep and gut health. They demonstrated that severe sleep loss causes accumulation of ROS in the gut (but not the brain) of both flies and mice, which was associated with death of the flies (the short cycle of sleep restriction did not allow for it to be demonstrated also in mice). Importantly, all these effects could be prevented by oral administration of antioxidant compounds or through gut-targeted transgenic expression of antioxidant enzymes. Many people use caffeinated coffee to keep awake, and while caffeine may favor insomnia [211], the antioxidant components of coffee (like melatonin, one of the antioxidant compounds used in the mentioned study by Rogulja and collaborators, [225]) may prevent accumulation of ROS in the gut and avoid the deleterious effect of voluntary sleep restriction.

### 4.3. Aminoacids and Their Derived Hormones

One of the compounds found naturally in coffee is tryptophan (Trp), an essential amino acid that must be supplied in the diet. Tryptophan is absorbed using the sodium-dependent neutral amino acid transporter, sodium-dependent neutral amino acid transporter (B^0^AT-1), which needs to be stabilized through interaction with the angiotensin-converting enzyme 2 (ACE2) [226]. Tryptophan absorption results in the secretion of α-defensins, cysteine-rich cationic peptides with antibiotic activity against a wide range of bacteria and other microbes, making dietary Trp essential for gut microbiota homeostasis [227,228]. Importantly, the aberrant absorption of Trp (which may occur due to cell surface downregulation of ACE2 during chronic stress, [229], or infection by the severe acute respiratory syndrome coronavirus 2 (SARS-CoV-2) [230]) leads to manifestations of colitis, such as diarrhea [231]. This amino acid is also essential for vitamin B_3_ (niacin) synthesis, and the deficit of this vitamin causes pellagra, a disease characterized by diarrhea, inflammation, and protein malnutrition, with skin and CNS manifestations [232]. Importantly, recent studies also revealed that niacin deficiency might be associated with Alzheimer’s, Parkinson’s, and Huntington’s diseases; cognitive impairment; or schizophrenia [232].

Once Trp is consumed and absorbed from the gut, it is made available in the circulation (the majority bound to albumin) and crosses the BBB to be involved in serotonin synthesis in the CNS [233,234]. Serotonin is a neurotransmitter that regulates different physiological aspects, such as behavior, learning, appetite, and glucose homeostasis [235]. Five percent of total body serotonin is brain-derived [235], whereas the majority of serotonin (95%) is produced from Trp in the ECs of the gastrointestinal tract [233]. ECs act as a sensory transduction component in the gastrointestinal mucosa. Serotonin is released by ECs after food intake, intraluminal distension, or efferent vagal stimulation, and its primary targets are the mucosal projections of primary afferent neurons including the vagal nerve [236]. Dietary and peripheral serotonin do not cross the BBB, implying that peripheral serotonin exerts different functions compared to brain-derived serotonin [235]. Peripheral serotonin is involved in the regulation of glucose and lipid homeostasis by acting on pancreatic β-cells, on hepatic cells, and on white adipocytes [235]. Serotonin is also involved in the regulation of visceral pain, secretion, and initiation of the peristaltic reflex, and altered levels of this hormone are also detected in many different psychiatric disorders. Symptoms of some gastrointestinal functional disorders may be due to deregulation of CNS activity, dysregulation at the peripheral level (intestine), or a combination of both (brain–gut axis) by means of neuro-endocrine-immune stimuli. In addition, several studies have demonstrated the profibrogenic role of serotonin in the liver, showing that it works synergistically with platelet-derived growth factor in stimulating hepatic stellate cell proliferation [237].

Another neurotransmitter synthesized from Trp in the brain is melatonin [238]. Melatonin has a crucial role in the control of the circadian cycle and it is also a powerful free radical scavenger and antioxidant [239]. Coffee is a source of melatonin, but the bioavailability of this compound in humans is low (around 3%) [240] and caffeine reduces endogenous nocturnal melatonin levels [238], with a significant impact on duration and quality of sleep [211].

GABA is a major inhibitory neurotransmitter of the CNS and is normally present in high concentrations in many brain regions. It is also found in green coffee beans (Table 1). Although the ability of GABA to cross the BBB is still unclear [241], its analgesic, anti-anxiety and hypotensive properties may be due to a local effect on gastrointestinal receptors, to circulating GABA, or to certain amount of GABA that might cross the BBB [196,242].

### 4.4. Maillard Reaction Products: Melanoidins

Dietary fiber and melanoidins, the latter also known as Maillardized dietary fiber [184], are likewise present in coffee (Table 1) and have health-promoting properties in the gut and possibly in the brain. Dietary melanoidins, similarly to fiber, escape gastrointestinal digestion, reach the colon, and become substrates for the gut microbiota [243]. In the gut, dietary fiber increases fecal bulk, contributes to normal bowel function and to accelerated intestinal transit [244]. Non-digestible carbohydrates are fermented by the microbiota into SCFAs, and these metabolites have been attributed several health effects [196]. Curiously, studies carried out in male Tsumura Suzuki obese diabetes (TSOD) mice, an accepted mouse model of metabolic syndrome, showed that caffeine and CGA improved plasma SCFA profile after 16 weeks of daily consumption of these compounds. However, in this study coffee had no effects probably because dietary fiber content in coffee composition differs by brand [245]. SCFAs influence gastrointestinal epithelial cell integrity, glucose homeostasis, lipid metabolism, appetite regulation, and immune function and are able to cross the BBB [246]. Interestingly, human studies have reported that dietary fiber can be isolated from SCGs and have chronobiotic effects [247], in addition to promoting short-term appetite and reducing energy consumption [248]. Moreover, a very recent randomized, crossover study of 14 healthy subjects reported that coffee melanoidins consumed at breakfast reduce daily energy intake and modulate postprandial glycemia and other biomarkers [249].

## 5. Conclusions

Coffee is a complex variable mixture of many compounds whose effects may vary according to their origin, processing, bioavailability, and possible synergistic and/or antagonistic effects.

Epidemiological studies have suggested that coffee brew may exert multiple effects on the digestive tract, including antioxidant, anti-inflammatory, and antiproliferative effects on the mucosa, and pro-motility effects on the muscle layers. However, in high contrast with what is known for other body systems and functions (i.e., cardiovascular system, CNS), the knowledge accumulated thus far regarding the effects of coffee and specific coffee-derived compounds on the gastrointestinal tract as a whole or on their different organs, as well as the specific mechanisms of action exerted on the different cell types present in the gut wall along the whole system, is strikingly scarce, despite the fact that the gastrointestinal tract is the first body system that comes in contact with ingested coffee. Furthermore, the impact of coffee and its derivatives on brain–gut axis health (from emotions to neurodegeneration) has only recently been addressed.

Coffee is recognized as one of the most popular beverages and largest traded produces worldwide, with millions of people consuming it on a daily basis [250]. Furthermore, the coffee plant *Coffee* sp. offers much more than the traditional beverage, and its by-products, including coffee flowers, leaves, pulp, husk, parchment, green coffee, silver skin, and SCGs, have become an attractive potential source of ingredients for new functional foods [251].

Hopefully, the currently huge interest in coffee and coffee by-products will help obtain robust scientific evidence clarifying their impact and mechanisms of action underlying their health-promoting properties in the gastrointestinal tract. Moreover, it is possible that targeted functional foods will soon be developed to specifically protect or improve gastrointestinal and brain–gut axis health.

## Figures and Tables

**Figure 1 nutrients-13-00088-f001:**
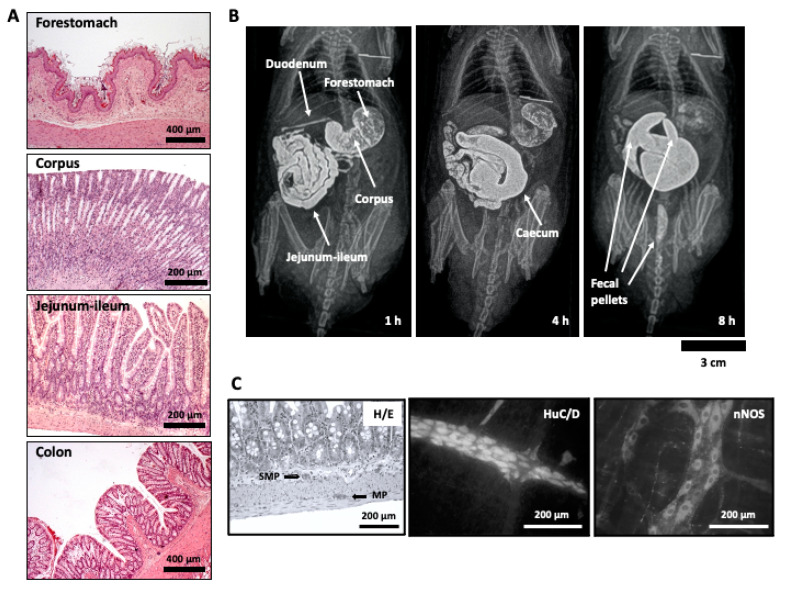
(**A**) Histological appearance of the wall of forestomach, corpus, jejunum-ileum (the longest part of the small intestine), and colon. (**B**) Organs of the rat gastrointestinal tract visualized by radiographic methods at different time points after intragastric barium administration in a conscious rat. Since rats do not vomit, barium can only progress towards the anus: 1 h after contrast, the two parts of the rat stomach (forestomach and corpus) can be distinguished, as well as the duodenum and the jejunum-ileum; 4 h after contrast, the stomach and small intestine can still be partially seen but now the cecum is filled with contrast; 8 h after contrast, the stomach and small intestine are barely seen, but the cecum is well filled with contrast and some fecal pellets are present within the colon. (**C**) Microscopic images showing the appearance of the enteric nervous system: left, location of the submucous (SMP) and the myenteric plexuses (MP) within the rat ileal wall in histological sections, stained with hematoxylin/eosin (H/E); middle and right, whole-mount or “sheet-like” preparations (from guinea pig ileum), obtained after dissecting away mucosa, submucosa, and circular muscle, leaving behind only the longitudinal muscle layer with the myenteric plexus attached; whole-mount preparations were processed immunohistochemically to show all the neurons with the pan-neuronal marker HuC/D (middle), or the specific subpopulation of neurons immunoreactive to neuronal nitric oxide synthase (nNOS), for which both somata and nerve fibers, but not nuclei, can be distinguished (right).

**Figure 2 nutrients-13-00088-f002:**
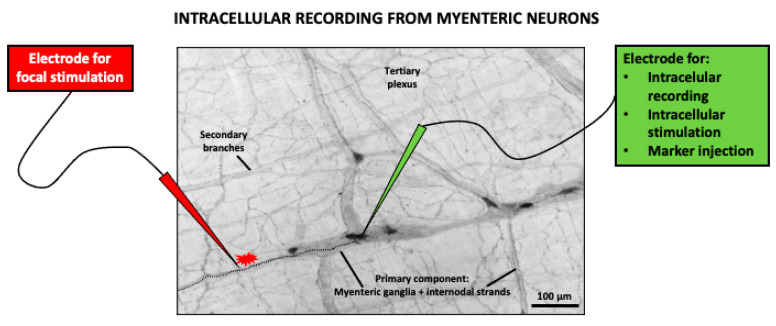
Intracellular recording from myenteric neurons. A fixed whole-mount preparation, processed immunohistochemically to show calretinin positive neurons, is used to illustrate how electrical activity of myenteric neurons would be recorded using current clamp electrophysiological modality. Calretinin immunoreactivity in guinea pig ileum whole-mount preparations allows one to distinguish the different components of the myenteric plexus: the primary component, which includes the myenteric ganglia and the internodal strands; the secondary branches that run circumferentially; and the tertiary plexus, a web of fine nerves that correspond to axons derived from excitatory longitudinal muscle motor neurons [165]. The intracellular recording electrode is represented in green (right)—this electrode allows for the recording of neuronal electrical activity, as well as direct intracellular stimulation of the cell with depolarizing or hyperpolarizing continuous or pulsed current, and marker injection to allow the impaled neuron to be visualized after immunohistochemical processing. The electrode for focal, extracellular stimulation is represented in red (left); this is placed on top of a circumferential internodal nerve strand. If the strand carries an axon (represented as a dotted line) that synapses on the impaled neuron, then focal stimuli (represented as a red blast symbol) will cause neurotransmitter release from the axon terminal and a postsynaptic potential on the impaled neuron (see Figure 3 for morphological and electrophysiological classifications of myenteric neurons).

**Figure 3 nutrients-13-00088-f003:**
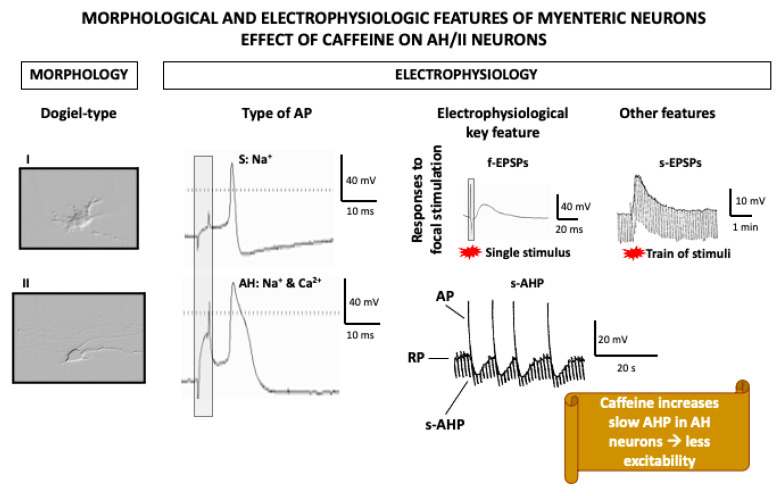
Morphological and electrophysiologic features of myenteric neurons and effect of caffeine on AH/II neurons. By use of intracellular recording methods illustrated in Figure 2, two main classes of myenteric neurons can be distinguished. According to morphology (left), neurons are classified as Dogiel type I (top) or Dogiel type II (bottom). These neurons broadly correspond to electrophysiological types S and AH, respectively. S neurons display short sodium-dependent APs, whereas APs of AH neurons are wider and depend on the entry of both Na^+^ and Ca^2+^, displaying a “hump” during the falling phase of the AP, due to Ca^2+^ entry. S neurons respond to single focal electrical stimuli with fast excitatory postsynaptic potentials (f-EPSPs), which are not seen in AH neurons, although both classes may respond to trains of focal stimulation with slow excitatory postsynaptic potentials (s-EPSPs). Finally, AH neurons display a s-AHP, due to K^+^ efflux dependent on the increase in intracellular free Ca^2+^ ([Ca^2+^]i) released from ryanodine-dependent stores. This s-AHP is increased and prolonged by caffeine, making these neurons, which are intrinsic peripheral nerve afferents, less excitable. Abbreviations: AP, action potential; f-EPSP, fast excitatory postsynaptic potential; RP, resting potential; s-AHP, slow after hyperpolarization; s-EPSP, slow excitatory postsynaptic potentials. Light grey blocks with dotted border, artifact of the stimulus; red blast symbol, focal stimulus.

**Figure 4 nutrients-13-00088-f004:**
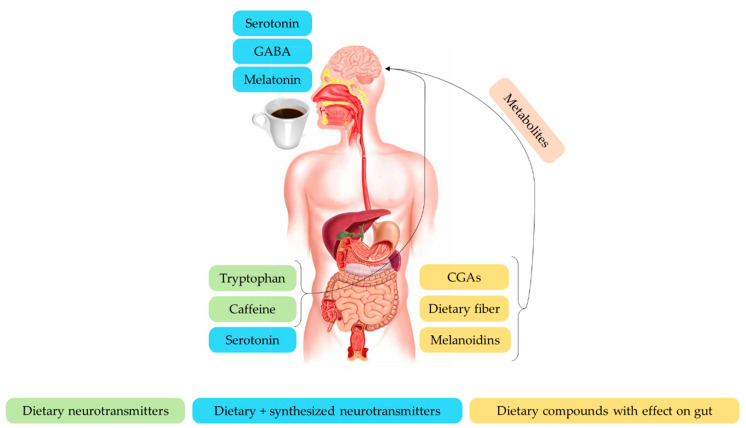
Effect of coffee compounds on the brain–gut axis. Abbreviations: CGAs, chlorogenic acids; GABA, γ-amino butyric acid.

**Table 1 nutrients-13-00088-t001:** Chemical composition of Arabica green, roasted, filtered, and cold brew coffee.

Constituent	Green Coffee Beans (100 g)	Roasted Coffee Beans (100 g)	Filtered Coffee Brew (330 mL)	Cold Brew Coffee (330 mL)
Carbohydrates	9–12.5 g	38 g	0 g	0.1 g
Fiber	46–53 g	31–38 g	1.2 g	0 g
Lipids	15–18 g	17 g	0.1 g	0 g
Proteins	8.5–12 g	7.5–10 g	0.1 g	0.1 g
Free amino acids	0.2–0.8 g	ND	NR	NR
Tryptophan	0.14 g	NR	0.028 g	NR
GABA	0.11 g	NR	NR	NR
Caffeine	0.8–1.4 g	1.3 g	0.244 g	0.412 g
Melatonin	0.7 mg	0.9 mg	0.026 mg	NR
Serotonin	1.3 mg	0.9 mg	0.048 mg	NR
Trigonelline	0.6–2.0 g	1 g	0.026 g	NR
Chlorogenic acids	4.1–9.2 g	1.9–2.7 g	0.009 g	13.2 g
Melanoidins	0 g	23 g	0.6 g	NR
Acrylamide	0 μg	24.4 μg	0.6–8.5 μg	1.4–1.8 μg
Ash	3–5.4 g	4.5 g	0.1 g	0 g
References	[9,12,13,14,15,16]	[9,12,14,17]	[12,13,17,18,19,20,21,22,23,24,25,26]	[19,21,27,28,29]

GABA, γ-aminobutyric acid; ND, not detected; NR, not reported.

## Data Availability

Data sharing not applicable.

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
