# Peer review of "Effects of Coffee and Its Components on the Gastrointestinal Tract and the Brain–Gut Axis"

_nutrients, 2020, doi:10.3390/nu13010088_

Round 1
Reviewer 1 Report
Coffee is a complex beverage with a variable mixture of many compounds and by-products whose effects may vary according to their origin, processing, bioavailability and possible synergistic and/or antagonistic effects.
In this manuscript, Dr Raquel Abalo and co-workers made an attempt to provide an overview of the effect of coffee, its components and its by-products on the gastrointestinal mucosa and the Brain-Gut Axis.
The overall representation of the manuscript is comprehensively described. Appropriate and adequate references to related and previous work are cited and clearly presented. However, the authors still need to make a minor correction.
Particular comments:
- Some important references have been omitted (Sci Rep., 2018, 8, 16173; Eur J Cancer 2010, 46,1873-1881).
- Multiple places are identified that the manuscript format is not under this Nutrients article’s guidance for authors. The format of all cited references is totally wrong since no DOI is needed. The senior author needs to proofread the whole manuscript with great caution.
Author Response
Coffee is a complex beverage with a variable mixture of many compounds and by-products whose effects may vary according to their origin, processing, bioavailability and possible synergistic and/or antagonistic effects.
In this manuscript, Dr Raquel Abalo and co-workers made an attempt to provide an overview of the effect of coffee, its components and its by-products on the gastrointestinal mucosa and the Brain-Gut Axis.
The overall representation of the manuscript is comprehensively described. Appropriate and adequate references to related and previous work are cited and clearly presented. However, the authors still need to make a minor correction.
ANSWER: We would like to thank the reviewer for the positive consideration towards our manuscript. We have tried to address his comments as shown below.
Particular comments:
- Some important references have been omitted (Sci Rep., 2018, 8, 16173; Eur J Cancer 2010, 46,1873-1881).
ANSWER: Thank you for your suggestions. We have added the references. Please see first paragraph in section 2 (revised and Cancer 2010 is included) and section 4.4. (revised and Sci. Rep. 2018 is included). In addition, we have included a new paragraph leading with the effects that sleep and antioxidants (including those present in coffee) have on gut health and their relationship with death promoted by sleep deprivation (and rescue promoted by antioxidants). We hope the new version of our manuscript is now acceptable for publication.
- Multiple places are identified that the manuscript format is not under this Nutrients article’s guidance for authors. The format of all cited references is totally wrong since no DOI is needed. The senior author needs to proofread the whole manuscript with great caution.
ANSWER: Thank you very much for your comment. We have reviewed the manuscript and the DOI numbers were removed from each reference.
Reviewer 2 Report
Manuscript ID: nutrients-1050845
Title: Effects of Coffee and its Components on the Gastrointestinal Tract and the Brain-Gut Axis
This manuscript is also a good brief updated review on the bioactive constituents of coffee and their effects not only on the brain-gut axis, but also in other organs and systems.
This manuscript is very well written and documented. A good source of information to whoever is interested in an update or a better understanding of the effects of coffee in human health.
We need more papers like this, specially covering areas with a lack of studies, such as Gut-brain axis and the effects of coffee daily consumption. I congratulate the group of researchers involved in this extensive review of the effect of coffee and its bioactive components. It was a delightful and very interesting reading.
Author Response
We need more papers like this, specially covering areas with a lack of studies, such as Gut-brain axis and the effects of coffee daily consumption. I congratulate the group of researchers involved in this extensive review of the effect of coffee and its bioactive components. It was a delightful and very interesting reading.
ANSWER: We would like to thank the reviewer for the positive consideration towards our manuscript